# The Role of NKG2D and Its Ligands in Autoimmune Diseases: New Targets for Immunotherapy

**DOI:** 10.3390/ijms242417545

**Published:** 2023-12-16

**Authors:** Leiyan Wei, Zhiqing Xiang, Yizhou Zou

**Affiliations:** Department of Immunology, School of Basic Medical, Central South University, Changsha 410083, China; 216511057@csu.edu.cn (L.W.); 216511061@csu.edu.cn (Z.X.)

**Keywords:** autoimmune disease, natural killer group 2 member D, rheumatoid arthritis, multiple sclerosis, inflammatory bowel disease, celiac disease

## Abstract

Natural killer (NK) cells and CD8^+^ T cells can clear infected and transformed cells and generate tolerance to themselves, which also prevents autoimmune diseases. Natural killer group 2 member D (NKG2D) is an important activating immune receptor that is expressed on NK cells, CD8^+^ T cells, γδ T cells, and a very small percentage of CD4^+^ T cells. In contrast, the NKG2D ligand (NKG2D-L) is generally not expressed on normal cells but is overexpressed under stress. Thus, the inappropriate expression of NKG2D-L leads to the activation of self-reactive effector cells, which can trigger or exacerbate autoimmunity. In this review, we discuss the role of NKG2D and NKG2D-L in systemic lupus erythematosus (SLE), rheumatoid arthritis (RA), multiple sclerosis (MS), type I diabetes (T1DM), inflammatory bowel disease (IBD), and celiac disease (CeD). The data suggest that NKG2D and NKG2D-L play a pathogenic role in some autoimmune diseases. Therefore, the development of strategies to block the interaction of NKG2D and NKG2D-L may have therapeutic effects in some autoimmune diseases.

## 1. Introduction

Autoimmune diseases result in damage to and the destruction of host tissues due to the misrecognition of self-antigens by the immune system. These diseases affect different organs and systems, such as the joints, nervous system, and muscles [1]. Natural killer (NK) cells and T cells are two important types of immune cells of innate and adaptive immunity, respectively. The responses of NK cells and T cells to pathogens and tumors are regulated by signals from a variety of receptors expressed on their cell surfaces that can initiate, enhance, or inhibit the function of their effector cells [2]. The recognition and activation of T cells are dominated by the antigen-specific T cell receptor (TCR), produced by somatic cell gene recombination and co-stimulatory molecules on their surfaces [3]. Meanwhile, NK cells are mainly determined by activating and inhibitory receptors on their surfaces [4]. Most NK receptors are also expressed on CD8^+^ T cells or CD4^+^ T cells, providing the costimulatory signal needed for activation, which also lowers the threshold for TCR activation by specific antigens [5]. Therefore, the abnormal expression of ligands of NK cell receptors on target cells may induce NK- or T-cell-mediated autoimmune responses [6]. Natural killer group 2 member D (NKG2D) is one of the most characterized receptors shared by both NK cells and T cells [7]. Here, we summarize all the data for NKG2D and the NKG2D ligand (NKG2D-L), which strongly support the idea that NKG2D and NKG2D-L are involved in the development of autoimmune diseases.

## 2. NKG2D and NKG2D-L

### 2.1. NKG2D Receptor

The NKG2D receptor consists of two disulfide-linked type II transmembrane glycoproteins whose extracellular region contains a C-type lectin-like structural domain [8]. The human NKG2D receptor is encoded by the killer cell lectin-like receptor K subfamily member 1 gene (*Klrk1*), which is localized to the NK gene complex on chromosome 12, namely 12p13.2. There are killer cell lectin-like receptor D1 (*KLRD1*) (CD94) gene clusters on the centromeric side and killer cell lectin-like receptor C4 (*KLRC4*) (NKG2F), killer cell lectin-like receptor C3 (*KLRC3*) (NKG2E), killer cell lectin-like receptor C2 (*KLRC2*) (NKG2C), and killer cell lectin-like receptor C1 (*KLRC1*) (NKG2A) gene clusters on the telomere side, as shown in Figure 1 [9]. The mouse homologous gene *Klrk1* exists on mouse chromosome 6 and also has limited polymorphisms [10]. The homologous gene of *Klrk1* is present in the genomes of all mammals, indicating that the gene is highly conserved in mammals.

With the exception of NKG2D, which is a homodimer receptor, all members of the NKG2 family form heterodimeric receptors with CD94 [11]. NKG2D is a multi-subunit receptor complex in which NKG2D signaling is mediated by specialized signal junctions [12]. Mouse NKG2D can bind to two different adaptors, DNAX activating protein 10 (DAP10) and DNAX activating protein 12 (DAP12), while human NKG2D uses only DAP10 [13]. The alternative splicing of mouse NKG2D produces two different transcripts [13]: NKG2D-long, which is constitutionally expressed on NK cells and is only related to DAP10, and NKG2D-short, initially expressed only on activated NK cells and later found to be expressed in naive mouse NK cells [14]—it is related to DAP10 or DAP12. DAP10 has a YXXM (Tyr-XX-Meth) sequence in the cytoplasm, which functions to recruit the P85 subunit of phosphatidylinositol 3 kinase (PI3K) and growth factor receptor binding protein 2 (Grb2) to activate the PI3K signaling pathway and Vav1-SOS signaling pathway, respectively [15,16,17]. DAP12 contains an immunoreceptor tyrosine-based activation motif (ITAM), whose phosphorylation leads to the recruitment of zeta chain-related protein kinase 70 (ZAP70) and splenic tyrosine kinase (Syk) [18]. Each NKG2D homodimer binds to two homodimers of DAP10 to form a hexameric complex [10]. Mouse immune cells express both the NKG2D-long and NKG2D-short subtypes, and NKG2D can bind to DAP10 and DAP12 [13]. Humans express only the NKG2D-long isoform, whose NKG2D receptor can only bind to DAP10 to form the NKG2D complex, as shown in Figure 2 [19].

NKG2D is expressed in NK cells, γδT lymphocytes, CD8^+^ T lymphocytes, NKT cells, and some CD4^+^ T cells [5]. NKG2D expression is significantly different between humans and mice. At rest, mouse CD8^+^T cells do not express NKG2D, but all human peripheral blood CD8^+^ T cells express this receptor, including cells expressing CD28^−^ [20,21]. In addition, NKG2D is expressed in the vast majority of human peripheral blood γδT cells [20], whereas it is selectively expressed in mouse γδT cells [22]. For example, mouse intestinal epithelial γδT cells do not express NKG2D, but human intestinal epithelial γδT cells all express low levels of NKG2D and can increase their expression in response to interleukin-15 (IL-15) stimulation [23]. In addition, human CD4^+^ T cells, similarly to mouse CD4^+^ T lymphocytes, generally do not express NKG2D [20], but the upregulation of NKG2D expression has been observed in T cell subsets in patients with certain autoimmune diseases.

### 2.2. NKG2D-L

Humans have two families of NKG2D-L, the MHC class I chain-related protein (MIC) and UL16-binding protein (ULBP) families [24]. The *MIC* gene family consists of seven members (*MICA-MICG*), and only MHC class I chain-related proteins A (*MICA*) and B (*MICB*) encode functional transcripts [24]. The ULBP family consists of ten genes *RAET1E-N*, of which only six encode functional proteins (called ULBP1 (RAET1I), ULBP2 (RAET1H), ULBP3 (RAET1N), ULBP4 (RAET1E), ULBP5 (RAET1G), and ULBP6 (RAET1L)) [25]. *Mouse* NKG2D-L are composed of members of the murine UL-16-binding protein-like transcript 1 (Mult-1), RAE-1α-ε, and H60a-c glycoprotein families [24]. Similarly to major histocompatibility complex class I (MHC-I) molecules, MICs have α1-, α2-, and α3 extracellular domains, transmembrane (TM) domains, and cytoplasmic domains, but they do not bind to β2-microglobulins and do not present antigenic peptides [26,27]. The remaining human and mouse ligands are structurally similar to MICs, but they lack the α3 extracellular domain [24]. In addition, NKG2D-Ls are attached to the membrane in different ways, with some being transmembrane proteins, such as MICs, ULPB4, ULBP5, Mult-1, H60a, and H60b, while others are anchored proteins such as ULBP1–3, ULBP6, H60c, and Rae-1, attached to the membrane by glycosylphosphatidylinositol (GPI) anchors, as shown in Figure 2 [13]. These ligands are highly polymorphic, especially MICA and MICB, and 531 *MICA* alleles encoding 280 protein variants and 244 *MICB* alleles encoding 47 protein variants have now been identified in humans (https://www.ebi.ac.uk/ipd/imgt/hla/alignment/) (accessed on 1 November 2023). Studies have shown that NKG2D binding to various ligands generally shows higher affinity than many immune receptor–ligand interactions [26]. The dissociation binding constant ranges from ~1 × 10^−6^ M to 4 × 10^−9^ M [28,29,30,31]. The crystal structure of the NKG2D-NKG2D-L complex suggests the rigid adaptation and structural plasticity of the NKG2D receptor [32]. This is the reason that the same receptor can recognize multiple ligands [33].

NKG2D-Ls are stress proteins that generally show low expression on normal cells and prevent the development of autoimmune diseases, except in the presence of cytokines, viral infections, oxidative stress, ionizing radiation, and DNA damage, where their expression is increased [34]. NKG2D-Ls play an important immunosurveillance role in the immune system, aiming to remove transformed cells [9].

NK cells play an important role in both autoimmune diseases and cancer. Interactions between NK cell receptors and target cells influence disease progression. Moreover, pro-inflammatory cytokine production mediated by the activating receptor NKG2D is one of the major causes of disease outbreaks. It was found that the expression of activated NK cell receptors such as NKp30, NKG2D, DNAX accessory molecule-1 (DNAM-1), and CD16 was reduced in cancers, while the expression of inhibitory receptors such as NKG2A was increased in cancers such as breast cancer [35]. Furthermore, high expression of NKG2D-L was associated with a good prognosis and improved tumor invasion in cancer patients [36,37]. However, some tumors use a variety of mechanisms to reduce NKG2D-L expression levels in order to evade NKG2D-mediated immune surveillance. It was found that NKG2D-L could be cleaved to soluble NKG2D-L (sNKG2DL) by certain metalloproteinases [38]. In contrast, the binding of soluble MICs (sMIC) and NKG2D leads to the degradation of NKG2D through endocytosis, thus allowing tumor cells to escape immune surveillance by NKG2D [39]. sNKG2DL can also affect the recognition of pathogens by NK cells, reduce the cytotoxicity of NK cells, and inhibit the immune surveillance function of NK cells [40]. However, in autoimmune diseases, NK cells may enhance the immune response due to the production of cytokines that regulate the immune response [41]. A study by Schepis et al. in nephritis, rheumatoid arthritis (RA), and systemic lupus erythematosus (SLE) patients and healthy individuals showed that NK cells stimulated antibody production and exacerbated the disease in patients with SLE [42]. Moreover, in some autoimmune diseases, the presence of a large number of sNKG2DL failed to induce the downregulation of NKG2D, which could be attributed to its rich cytokine effects [43]. Furthermore, in tissue samples from patients with autoimmune diseases as well as in vivo experimental models, it was found that NKG2D-L upregulation and NKG2D^+^ lymphocytes were involved in their pathogenesis.

## 3. Role of NKG2D/NKG2D-L in Autoimmune Diseases

### 3.1. SLE

SLE is an autoimmune disease of unknown etiology that affects multiple systems, primarily the skin, joints, kidneys, and central nervous system (CNS), and is caused by a combination of genetic and environmental factors. SLE produces antibodies against its own antigens, forms immune complexes, and activates complements [44,45]. In the study by Dai et al. in adolescent SLE patients, increased frequencies of soluble MICA (sMICA), soluble MICB (sMICB), and interleukin-10 (IL-10)-producing NKG2D^+^CD4^+^ T cells were observed and were negatively correlated with the severity of SLE [46]. Plasma concentrations of sMICB in SLE patients have been reported to promote the expansion of NKG2D^+^CD4^+^ T cell subsets [47]. When investigating the mechanism of NKG2D^+^CD4^+^T cells in SLE, Yang et al. [48] found that NKG2D expression could be induced on normal CD4^+^ T cells by co-culturing them with monocytes from SLE patients; the induced NKG2D^+^CD4^+^ T cells were involved in the pathogenesis of SLE in the form of NKG2D-MIC interactions with CD14^+^ monocytes. This is in contrast to the findings of Dai et al. NKG2D^+^CD4^+^ T cells secrete interferons (IFN), tumor necrosis factor-α (TNF-α), and granzymes and exhibit direct cytotoxicity and cytolytic properties [33]. The treatment of MRL/Lpr mice with anti-NKG2D or anti-IFN-α receptor antibodies was shown to restore the number of Treg cells and significantly improve the symptoms of lupus disease. The mechanism may involve NKG2D^+^CD4^+^ T cells killing Treg cells in an NKG2D-NKG2D-L-dependent manner, thus participating in the pathogenesis of SLE [49].

Studies have shown that low numbers and low toxicity of NK cells are detected in the peripheral blood of SLE patients, and they are associated with the recurrence and disease activity of SLE [50]. NKG2D expression on the surfaces of NK cells was reduced in SLE patients compared to healthy controls [51], and there was a significant negative correlation between the expression of this receptor and SLE Disease Activity Index (SLEDAI) scores, disproving the previous conclusion [52] that “NKG2D is unlikely to play a key role in the pathogenesis of lupus” [53].

Many studies have shown that SLE has a strong genetic background. A Polish population study found that the *NKG2D 72Thr* gene variant protects against SLE [54]. Another study showed that SLE is associated with the single nucleotide polymorphism (SNP) rs2255336 of NKG2D [55]. In addition, the MICA polymorphism is significantly correlated with the incidence of SLE. For example, the *MICA 129Met* allele, *TMA9* allele, and *129Met/Met* genotype are positively correlated with SLE, while the *MICA 129Val* allele is negatively correlated with SLE [56]. The *MICB*009N* allele may be a risk factor for SLE, while the *MICB*014*, *MICA*010*, and *MICB*002* alleles are protective factors for SLE [57].

### 3.2. RA

RA is a chronic inflammatory disease characterized by joint inflammation, hyperplasia and swelling, the production of autoantibodies, and bone destruction [58]. Studies have shown that the severity of RA is associated with a large number of CD4^+^CD28^−^ T cells [43,48,59]. Studies have found that a large amount of TNF-α and IL-15 exists in the peripheral blood and synovial tissue of RA patients [43], and these cytokines can induce the expression of NKG2D in CD4^+^CD28^−^ T cells in these fluids [43,59,60,61,62,63]. In addition, the peripheral blood of RA patients contains large amounts of synoviocyte-derived sMICA [43,64]. In tumor patients, sMICA/sMICB reduces the expression of NKG2D on the surfaces of NK cells and weakens the function of NK cells, allowing tumor cells to evade NKG2D-mediated immune surveillance [13]. However, a decrease in NKG2D on the surfaces of CD4^+^CD28^−^ T cells in RA patients cannot be induced by sMICA, which may be related to the abundance of IL-15 in RA patients, as shown in Figure 3 [43].

Studies have shown that NK cell activity is significantly decreased in RA patients, which may be related to the presence of large amounts of interleukin-6 (IL-6), TNF-α, and interleukin-18 (IL-18) in their serum, which can reduce the expression of NKG2D on their surfaces [65]. A study of the interactions between fibroblast-like synovial cells (FLS) and NK cell lines (Nishi) in RA found that FLS express many ligands of NK cells and stimulate the degranulation of Nishi cells, and NKG2D is one of the key activating receptors involved in Nishi’s degranulation of FLS [66].

Although regulatory T cells (Tregs) show great promise in the treatment of RA, the small number of Tregs limits their further clinical use [67]. A study of RA patients refractory to anti-TNF-α therapy found that highly differentiated antigen epitope (AE)-specific CD8^+^ Teff cells were completely uninhibited by Tregs due to a mechanism that may be the result of AE-CD8^+^ Teff cells expressing an effector cell phenotype and gene profile and killing Treg cells in an NKG2D-dependent manner in vitro after antigen-specific activation [68]. Studies have shown an abundance of CD38^+^ NK cells in RA [69]. Whereas anthocyanin-3-O-glucoside (C3G) is an inhibitor of CD38, WangH found that C3A has a therapeutic effect on RA by exploring the effect of C3G on RA. Its mechanism of action may involve increasing the expression of Sirtuin6 (Sirt6) to inhibit the expression of NKG2D, decreasing the proportion of CD38^+^ NK cells, decreasing the secretion of pro-inflammatory cytokines, and increasing the proportion of Treg cells [69].

In addition to the human leukocyte antigen *(HLA)-DRB1 SE* allele, *MICA* and *NKG2D* polymorphisms are also susceptibility genes for RA [70]. A study of *MICA* polymorphisms and susceptibility to RA in Caucasian populations from France and Germany found that *MICA-250* (rs1051794) is associated with RA and independent of known HLA-DRB1 risk alleles, suggesting that *MICA* is a susceptibility gene for RA [71]. In addition, the *MICA-129 Val/Val* genotype was found to be associated with high levels of sMICA and increased the severity of RA in Tamils in South India [72]. Polymorphisms in NKG2D can alter the risk and severity of RA [73]. In addition, the NKG2D polymorphism also affects the responsiveness of RA patients to TNF inhibitors, and patients with the rs2255336 or rs1049174 heterozygous genotypes have a better European League Against Rheumatism (EULAR) response than those with homozygous genotypes [74].

### 3.3. Multiple Sclerosis (MS)

MS is a chronic inflammatory disease of the CNS with pathological features including demyelinating areas, neuron/axon loss, and glial cell proliferation [75]. Immune cells that have infiltrated into the CNS are associated with demyelination and neurodegeneration in MS [76]. Studies have shown that NKG2D^+^CD4^+^ T cells are associated with inflammatory CNS lesions [77]. The crossing of the blood–brain barrier by CD4^+^ T cells is an important step in the pathogenesis of MS. A large number of NKG2D^+^CD4^+^ T cells were found in the cerebrospinal fluid (CSF) of MS patients. In experimental autoimmune encephalomyelitis (EAE) animal models, the blocking of NKG2D inhibited the migration of NKG2D^+^CD4^+^ T cells to the CNS and weakened the killing effect on mouse oligodendrocytes, suggesting that NKG2D promoted the migration of NKG2D^+^CD4^+^ T cells across the blood–brain barrier [78]. The study found that IL-15 was significantly increased in the serum and CSF of MS patients, and it was mainly secreted by astrocytes and infiltrating macrophages. IL-15 can promote CD4^+^CD28^−^ T cells to express NKG2D and secrete perforin and granzyme B [77]. NKG2D^+^CD8^+^ T cells are also involved in the pathogenesis of MS and are located near cells that express IL-15 [79]. IL-15 also activates CD8^+^ T cells, exacerbating tissue damage, as shown in Figure 3 [79].

In women with MS, the NK cells are reversed postpartum [80]. Pregnancy was found to promote the marked transformation of NK cells into a regulatory CD56^bright^NK cell population that expressed receptors associated with cytotoxicity (such as the CD16^+^NKp46^high^ NKG2D^high^ NKG2A^high^ phenotype) [80]. ULBP4 was found to be predominantly expressed by astrocytes rather than neurons [81]. Soluble ULBP4 (sULBP4) was significantly elevated in the CSF of female MS patients compared to controls and male MS patients [81]. sULBP4 can affect the function of CD8^+^ T cells, such as enhancing the production of pro-inflammatory cytokines, granulocyte-macrophage colony-stimulating factor (GM-CSF), and interferon-γ (IFN-γ) and promoting the motor ability of CD8^+^ T cells [81], and this assertion has been confirmed in EAE models [82]. When NKG2D was blocked, the number and motility of CD8^+^ T cells co-cultured with astrocytes expressing NKG2D-L were increased, suggesting that NKG2D was involved in the interaction between CD8^+^ T cells and astrocytes [83]. In the treatment of MS patients, different drugs are used, such as interferon-β (IFN-β) and Fingolimod [84,85,86]. Patients with relapsing multiple sclerosis (RR-MS) under IFN-β treatment were found to have significantly increased levels of both NKG2D, an activating receptor on the surfaces of NK cells, and interleukin-22 (IL-22), suggesting that IFN-β treatment directs NK cells toward a pro-inflammatory state [84]. Moreover, Fingolimod treatment can cause the enrichment of the NK cell subpopulation defined by CD56^dim^CD16^++^KIR^+/−^NKG2A^−^CD94^−^CCR7^+/−^CX3CR1^+/−^NKG2C^−^NKG2D^+^NKp46^−^DNAM1^++^CD127^+^, which is characterized by aging. This also limits the activity of anti-microbial and anti-tumor NK cells in patients treated with Fingolimod [86]. In addition, studies have shown that CD56^dim^ NK cells in MS patients treated with both IFN-β1 and Fingolimod maintain functional responsiveness but show different transcriptomic signatures [85].

### 3.4. Type I Diabetes (T1DM)

T1DM is a chronic autoimmune disease that is characterized by the T-cell-mediated destruction of insulin-producing β cells in the pancreatic islets, leading to a decrease in insulin levels in the body and ultimately leading to hyperglycemia in patients [87]. Studies have shown that NKG2D plays an important role in the development of T1DM [88]. In recent years, however, there have been conflicting conclusions about the role of NKG2D in the pathogenesis of T1DM [89,90,91,92,93]. Ogasawara found that diabetic pancreatic cells from NOD mice expressed RAE-1 and that CD8^+^ T cells infiltrating the pancreas expressed NKG2D. When NKG2D is blocked with antibodies, the function of CD8^+^ T cells is inhibited, which completely prevents the development of the disease [89]. This is consistent with Kjellev’s [91] conclusion that NKG2D plays a key pathogenic role in T1DM, but contrary to the conclusions of Rodacki and Van Belle’s studies. Rodacki found that, in patients with T1DM, the expression of NKG2D decreased only slightly and was independent of the duration of the disease [90]. Van Belle found that NKG2D expression was increased on CD4^+^ and CD8^+^ T cells in virus-induced diabetes and that when NKG2D was blocked with an antibody, it failed to reverse the recent onset of diabetes in NOD mice [92].

Hyperglycemia can inhibit the expression of NKG2D, NKp46, and granzyme B on NK cells, thus affecting the activity of NK cells [94]. Gluten has been found to affect the development of T1DM in NOD mice. Larsen [95] found that gluten-based food can increase NKG2D on the surfaces of NK cells, thereby increasing the activity of mouse NK cells against pancreatic β cells. A gluten-free diet downregulated NKG2D on infiltrating NK cells and CD8^+^ T cells in the pancreas in NOD mice, thereby reducing the incidence of T1DM [96]. Interleukin-12 (IL-12) and IL-18 can synergistically promote the inflammatory response of the Th1 type, which is considered to be a promoter of T1DM pathogenesis. In T1DM patients, serum IL-12 and IL-18 were significantly increased compared with a control group and were positively correlated with glycosylated hemoglobin A1c (HbA1c) levels [97]. Dean [98] found that IL-12 and IL-18 can synergistically activate NK cells, and the activated NK cells can increase the expression of NKG2D and CD25, which inhibits the regulatory function of Treg cells.

Tumor cells can secrete large amounts of sNKG2D-L to evade immune surveillance by NK cells. Blevins’ [88] team isolated NKG2D soluble ligand (sRAE-1) plasmid DNA and delivered a therapeutic plasmid targeting the pancreas, which reduced the interaction between β cells and infiltrating NKG2D-positive lymphocytes, thereby effectively protecting β cells from autoimmune destruction and preventing T1DM. Trembath [99] directly expressed NKG2D-L on β cells in the islets of NOD mice, where diabetes was suppressed and the number of central memory CD8^+^ T cells increased, suggesting a protective role of central memory CD8^+^ T cells in T1DM.

### 3.5. Inflammatory Bowel Disease (IBD)

IBD is a group of intestinal diseases that cause digestive tract inflammation over a long period of time, mainly manifested as Crohn’s disease (CD) and ulcerative colitis (UC) [100]. In general, CD is dominated by Th1 and Th17 cells, while UC is dominated by Th2 cells [101]. Biological analyses have shown that the mucosal Th1 cytokine TNF plays a partial role in regulating the Th2-dominated Th1/Th2 imbalance in UC compared to Th1-mediated CD [102]. Studies have found that MICA is significantly upregulated in the intestinal epithelial cells (IECs) of patients with CD and UC [103,104,105,106] and can participate in the pathogenesis of IBD through NKG2D–MICA interaction. Compared with the control group and UC patients, the expression of NKG2D in CD4^+^ cells was positively correlated with the occurrence of CD lamina propria lesions [107]. NKG2D^+^CD4^+^ T cells comprise most of the oligo-clones of CD mucosal T cells [108], which secrete the inflammatory cytokines TNF-α and interleukin-17 (IL-17) [109]. CD4^+^NKG2D^+^ T cells with a Th1-type cytokine profile are increased in the periphery and mucosa of CD and produce IFN-γ to kill MICA-expressing target cells through NKG2D–MICA interactions, as shown in Figure 4 [104]. Autologous mucosal T cells were found to directly induce epithelial cell death in CD patients but not in controls, but this effect was inhibited by blocking antibodies to CD103 and NKG2D [110].

Tumor necrosis factor (TNF)-like cytokine 1A (TL1A) is a pro-inflammatory cytokine that is ubiquitous in the gut. High concentrations of TL1A were present in both IBD patients and mouse models. TL1A knockout (KO) mice had reduced numbers of TCRγб^+^ and CD8^+^ T cells in the small intestinal epithelium, as well as decreased expression of NKG2D [111]. In dextran sulfate sodium (DSS)-induced colitis, splenic NKG2D^+^CD4^+^ T cells could be divided into two subpopulations based on the expression of NK1.1, namely TGF-β^+^FasL^+^T-bet^+^NK1.1^−^ cells and IFN-γ^+^IL-17^+^IL21^+^granzymeB^+^perforin^+^T-bet^−^RORγt^+^NK1.1^+^ cells. NK1.1^−^NKG2D^+^CD4^+^T cells delayed the onset of DSS-induced colitis, and their protective effect was dependent on transforming growth factor beta (TGF-β). In contrast, NK1.1^+^NKG2D^+^CD4^+^ T cells exacerbated the outcome of colitis [112]. Hosomi found that the deletion of the X-box binding protein 1 (*Xbp1*) gene in intestinal epithelial cells led to the increased expression of ULBP and spontaneous enteritis in mice, while blocking NKG2D inhibited the cytolysis of endoplasmic reticulum (ER) pressurized epithelial cells in vitro and spontaneous enteritis in vivo [113,114].

To assess the functionality of Tesnatilimab (NCT02877134), a monoclonal antibody-targeting NKG2D, Allez conducted a clinical study in patients with CD who had failed to respond to conventional therapies. The study showed that the Crohn’s Disease Activity Index (CDAI) scores of injected monoclonal antibodies were more altered from the baseline when 400 mg/kg monoclonal antibody or placebo was injected subcutaneously, but it did not reveal a dose-responsive signal for this monoclonal antibody [115]. However, in a randomized, double-blind, parallel-group, placebo-controlled trial (NCT01203631), the anti-NKG2D antibody NNC0142-0002 showed clinical efficacy in patients with CD, especially in patients treated with biologically based new drugs [116].

### 3.6. Celiac Disease (CeD)

CeD is a complex small intestinal disorder in which susceptible individuals expressing HLA-DQ2 or DQ8 molecules develop a Th1 immune response to gluten in wheat, barley, and rye, resulting in the loss of oral tolerance (LOT) to gluten, manifested by villous atrophy, crypt cell hyperplasia, and the infiltration of inflammatory cells in the lamina propria and epithelium [117,118,119]. Currently, the only treatment is to exclude gluten from the diet [120]. MICA/B expression in the intestinal mucosa of CeD patients is associated with the dysregulation of mucosal homeostasis [121]. During active CeD, IEC strongly expresses MIC molecules and shows high levels of IL-15 [122,123,124]. Hue [123] cultured intestinal biopsy tissues with gluten protein or gluten protein peptide in vitro and found that IEC strongly expressed MICA. The same induction effect was observed in IEC when recombinant IL-15 was substituted for gluten protein or gluten protein peptide. When IL-15 is blocked, the MICA-inducing effect of gluten is also blocked, suggesting that IL-15 plays a key role in the intestinal mucosal damage caused by gluten protein intake. In addition, IL-15 can induce the high expression of NKG2D in intraepithelial lymphocytes (IELs) [124], and its ligand MICA is strongly expressed on IECs, thus enabling IEL to kill IEC-expressing MICA in an NKG2D-mediated manner, independently of TCR [122]. IL-15 can also act as a stimulating molecule for NKG2D-mediated cell lysis, leading to the release of arachidonic acid, which in turn promotes the activation and recruitment of granulocytes, resulting in more intestinal inflammation. Compared to patients with active celiac disease (ACD), patients on a gluten-free diet had a higher frequency of IEL expression of the inhibitory receptors NKG2A and TGF-β1 by CD8^+^ TCRgammadelta^+^. When TGF-β1 is blocked alone or the binding of NKG2A and HLA-E is blocked simultaneously, CD8^+^ TCRgammadelta^+^ IELs can regulate CeD through their secretion of TGF-β1 [125].

CeD patients showed a unique gut microbiome composition and increased IgA response compared to healthy subjects. Especially at five years of age, twenty-six plasma metabolites, five cytokines, and one chemokine were significantly altered in patients with CeD. Among the twenty-six metabolites, there was a two-fold increase in taurodeoxycholic acid (TDCA). TDCA alone induced villus atrophy in C57BL/6J mice, increased the expression of NKG2D on the surfaces of CD4^+^T cells and NK cells, and decreased the proportion of Treg cells in IELs [126]. TDCA also reduced NK cell activation by downregulating NKG2D/NKp46 receptor expression in mouse splenocytes and male mouse PP. Moreover, in patients with active CeD, NKp44/NKp46 double-positive NK cells are significantly reduced [126,127]. Cytotoxic T-lymphocytes (CTLs) can produce and react to cysteinyl leukotrienes (Cyst-LTs) to kill target cells in a TCR-independent manner, a process that relies on NKG2D and IL-15. The mechanism may involve IL-15 collaborating with NKG2D to drive the upregulation of key enzymes related to Cyst-LT synthesis and the expression of Cyst-LT receptors. The blocking of the Cyst-LT receptor may be an effective strategy for the treatment of CeD or other conditions that may be related to NKG2D [128].

## 4. NKG2D and NKG2D-L Are Key Targets in Autoimmune Diseases

The expression of NKG2D is regulated by various cytokines, such as interleukin-2 (IL-2), interleukin-7 (IL-7), IL-12, IL-15, and IL-18, which can significantly upregulate the expression of NKG2D [129,130,131], whereas TGF-β, IFN-β1, and interleukin-21 (IL-21) downregulate NKG2D expression [130,132,133,134]. The current clinical trials being conducted for autoimmune diseases with NKG2D as the target are shown in Table 1. Five of the six clinical trials have been successfully completed and one was not pursued due to funding issues. Four of the five completed clinical trials have shown good results, particularly NCT02877134 [115] and NCT01203631 [135]. Animal experiments suggest that NKG2D is involved in the development of autoimmune diseases and blocking NKG2D-attenuated disease progression in certain colitis mice [91,136]. Treatment with NKG2D-neutralizing antibodies prevented the recipient rejection of parental BALB/c bone marrow and allowed the implantation of allogeneic BALB [137]. Treatment with a non-depleting anti-NKG2D monoclonal antibody in the pre-diabetic phase completely prevented disease by impairing the expansion of self-reactive CD8^+^ T cells [89]. NK cell-mediated heat shock protein 70 (HSP70)-pc-induced EAE tolerance involves the induction of H60 and its interaction with the NKG2D receptor. When H60 is blocked, HSP70-pc-induced EAE tolerance can be reversed [138]. In a colitis model, an anti-NKG2D antibody (CX5) significantly reduced disease progression in subjects with mild colitis, but did not reduce the disease severity in those with moderate-to-severe colitis [91]. After the transfer of CD4^+^CD45RB^High^ T cells into severe combined immunodeficiency (SCID) mice, treatment with a neutralizing anti-NKG2D MAb significantly inhibited marasmus associated with colitis, alleviated leukocyte infiltration, and reduced IFN-γ production by CD4^+^ T cells in the membranes propria. In patients with CD, a Phase IIa study with a blocking antibody against NKG2D showed a significant increase in clinical response after 12 weeks, suggesting that the interaction of NKG2D with its ligand is a viable therapeutic target [107]. It is worth considering that blocking NKG2D may inhibit interactions between immune and non-immune cells (including lymphoepithelial interactions) and reduce effector cell activity and function. NKG2D is also involved in the immune responses to various pathogens and tumors in the body. Therefore, blocking NKG2D decreases the anti-infection and anti-tumor functions of NK cells. Future trials should carefully select an appropriate dose range and consider how to assess the requirement for maintenance therapy after the initial induction period. These findings suggest that the NKG2D signaling pathway plays a key role in disease progression mediated by CD4^+^ T or CD8^+^ T cells and propose a new therapeutic target for autoimmune diseases.

## 5. Conclusions

Autoimmune diseases are fatal diseases mediated by immune cells. They are characterized by the loss of immune tolerance and by over-activated immune cells attacking healthy host cells, tissues, and organs, resulting in serious systemic or local organ damage and serious harm to patients’ health. To date, there are still no specific drugs for the treatment of autoimmune diseases, because the causes of autoimmune diseases remain uncertain. At present, the clinical treatment of autoimmune diseases mainly relies on glucocorticoids and immunosuppressants, but these drugs can have serious side effects, such as infection and tumors caused by low immune function. In recent years, many studies have suggested that NKG2D may play an important role in the development of autoimmune diseases. In autoimmune diseases such as MS, T1DM, and CeD, NKG2D can be induced to be expressed in the corresponding lymphocytes, so as to respond to the overexpression of NKG2D-L self-cells to produce a killing effect (as shown in Table 2). This killing effect can be independent of TCR. In addition, in SLE, RA, and IBD, activated NKG2D^+^CD4^+^T lymphocytes secrete several pro-inflammatory cytokines, such as GM-CSF, TNF-α, IL-17, and IFN-γ, which aggravate the destruction of their own cells or tissues. These studies suggest that the interaction of NKG2D and NKG2D-L can activate self-reactive T cells or NK cells, leading to the destruction of host tissues. Therefore, NKG2D and NKG2D-L are promising therapeutic targets. Given that the expression of NKG2D and NKG2D-L is regulated by a variety of factors, a number of targeted antibodies can be formulated to block the expression of NKG2D or NKG2D-L, thereby inhibiting the interaction; this may have important implications for the treatment of autoimmune diseases.

## Figures and Tables

**Figure 1 ijms-24-17545-f001:**
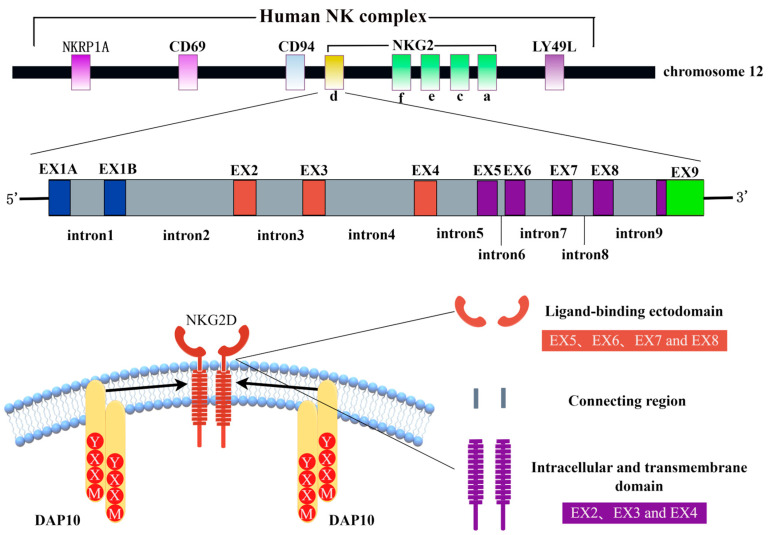
Schematic of NKG2D structure generated using Figdraw. Human NKG2D is located in the natural killer complex (NKC) of chromosome 12 (top). Human NKG2D consists of 10 exons and 9 introns (middle): exons 2–4 encode intracellular and transmembrane domains, and exons 5–8 encode ligand-binding domains (bottom). NKRP1A: natural killer cell surface protein P1A, NKG2: natural killer group 2, EX: exon, NKG2D: natural killer group 2 member D, DAP10: DNAX activating protein 10, and YXXM: Tyr-XX-Meth.

**Figure 2 ijms-24-17545-f002:**
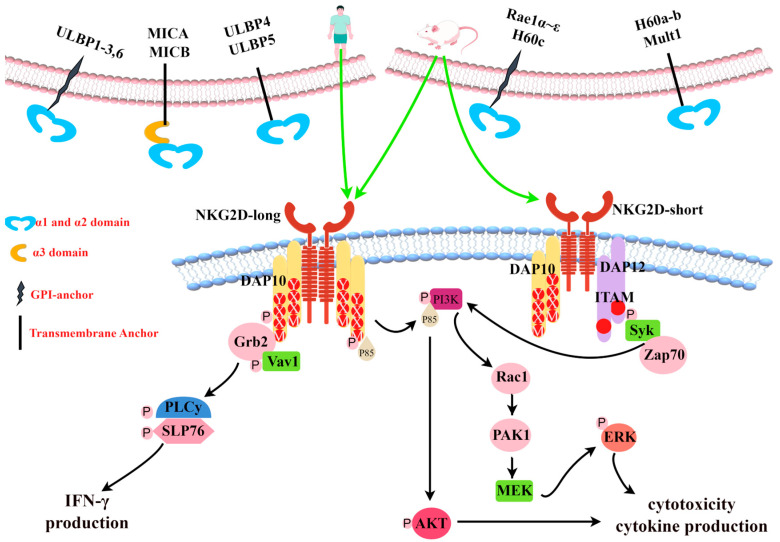
Diversity of NKG2D ligands and NKG2D signal transduction. Generated using Figdraw. All human and mouse NKG2D ligands known to date are shown (top). In mice, NKG2D exists as short (NKG2D-short) or long (NKG2D-long) splicing isomers. In humans, NKG2D is only expressed as NKG2D-long. Mouse NKG2D can bind to DAP10 or DAP12 signaling molecules, while human NKG2D only binds to DAP10. Pairing with DAP12 causes the phosphorylation of an activation motif (ITAM) based on the immune receptor tyrosine and triggers a Syk and/or Zap70 cascade. Pairing with DAP10 leads to tyrosine phosphorylation in the YINM group and triggers the PI3K and Grb2/Vav1 signaling cascade (bottom). ULBP: UL16-binding protein, MICA: MHC class I chain-related protein A, MICB: MHC class I chain-related protein B, Mult1: Murine UL-16-binding protein-like transcript 1, GPI: glycosylphosphatidylinositol, DAP10: DNAX activating protein 10, DAP12: DNAX activating protein 12, YXXM: Tyr-XX-Meth, ITAM: immunoreceptor tyrosine-based activation motif, Grb2: growth factor receptor binding protein 2, VaV1: Vav guanine nucleotide exchange factor 1, PLC: phospholipase C, IFN-γ: interferon-γ, PI3K: phosphatidylinositol 3 kinase, AKT: protein kinase B, Rac1: Ras-related C3 botulinum toxin substrate 1, PAK1: P21-activated kinase 1, MEK: mitogen-activated extracellular signal-regulated kinase, ERK: extracellular regulated protein kinases, Syk: splenic tyrosine kinase, and Zap70: zeta chain-related protein kinase 70.

**Figure 3 ijms-24-17545-f003:**
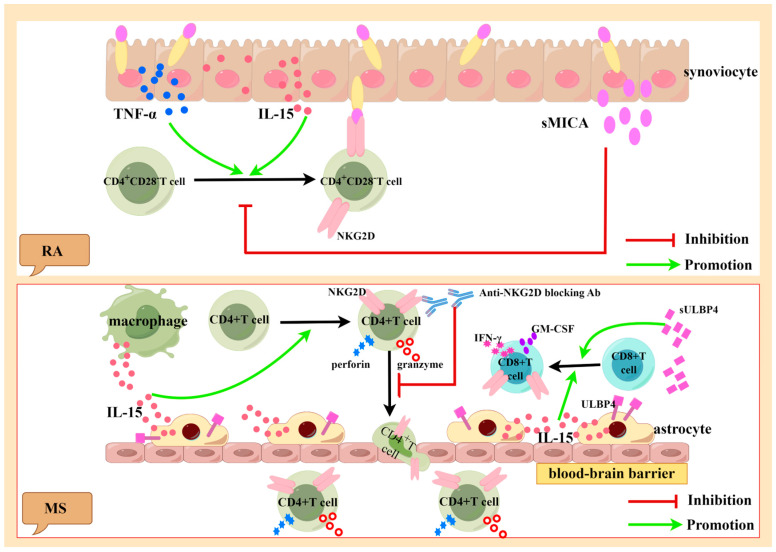
Generated using Figdraw. Pathogenic role of activated NKG2D-positive lymphocytes in RA and MS. Synovial cells of RA patients can express and secrete large amounts of TNF-α, IL-15, and sMICA. TNF-α and IL-15 can induce the expression of NKG2D in CD4^+^CD28^−^ T cells, which can kill the synovial cells expressing MICA. sMICA inhibits the expression of NKG2D in CD4^+^CD28^−^ T cells (above). Astrocytes and infiltrating macrophages can secrete large amounts of IL-15 and sULBP4. IL-15 can induce the expression of NKG2D in CD4^+^ T cells and CD8^+^ T cells. When NKG2D is blocked with antibodies, the migration of NKG2D^+^CD4^+^ T cells in the blood–brain barrier is inhibited. IL-15 can promote the expression of perforin and granzyme B in NKG2D^+^CD4^+^ T cells. sULBP4 promotes the CD8^+^T expression of pro-inflammatory factors GM-CSF and IFN-γ (below). TNF-α: tumor necrosis factor-α, IL-15: interleukin-15, sMICA: soluble MICA, NKG2D: natural killer group 2 member D, RA: rheumatoid arthritis, MS: multiple sclerosis, IFN-γ: interferon-γ, GM-CSF: granulocyte-macrophage colony-stimulating factor, ULBP4: UL16-binding protein4, and sULBP4: soluble ULBP4.

**Figure 4 ijms-24-17545-f004:**
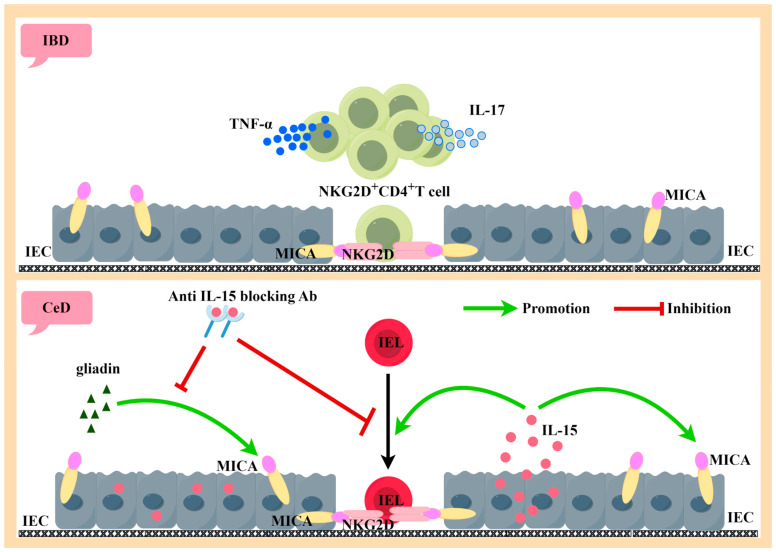
Generated using Figdraw. Pathogenic role of activated NKG2D-positive lymphocytes in IBD and CeD. NKG2D^+^CD4^+^ T cells are the majority of the oligo-clones of CD mucosal T cells, which secrete large amounts of the inflammatory cytokines TNF-α and IL-17. The interaction of NKG2D^+^CD4^+^ T cells with MICA-expressing intestinal epithelial cells has a killing effect on intestinal epithelial cells (above). IL-15 and MICA are highly expressed by IEC during active CeD. Both gluten protein and IL-15 can induce IEC expression of MICA. The inducible effect of gluten protein on MICA can be inhibited by neutralizing antibodies to IL-15. IL-15 can induce IEL to express NKG2D, thus producing an NKG2D–MICA killing effect with IEC-expressing MICA (below). IBD: inflammatory bowel disease, IEC: intestinal epithelial cells, MICA: MHC class I chain-related protein A, NKG2D: natural killer group 2 member D, TNF-α: tumor necrosis factor-α, IL-17: interleukin-17, CeD: celiac disease, IL-15: interleukin-15, and IEL: intraepithelial lymphocyte.

**Table 1 ijms-24-17545-t001:** Clinical trials targeting NKG2D for autoimmune diseases.

Title	Clinical Trial No.	Status	Actual Enrolment	Conditions	Phase	Year Last Updated
Predictive Factors of ANTI-TNF Response in Luminal Crohn’s Disease Complicated by Abscess	NCT02856763	Completed	125 participants	Crohn’s disease	/	20 January 2021
Safety and Efficacy Study of JNJ-64304500 in Participants With Moderately to Severely Active Crohn’s Disease	NCT02877134	Completed	388 participants	Crohn’s disease	Phase 2	17 February 2023
A Study of JNJ-64304500 as Add-on Therapy in Participants With Active Crohn’s Disease	NCT04655807	Withdrawn (sponsor decision)	/	Crohn’s disease	Phase 2	1 September 2021
Safety and Efficacy of NNC 0142-0000-0002 in Subjects With Moderately to Severely Active Crohn’s Disease	NCT01203631	Completed	78 participants	Inflammation Crohn’s disease	Phase 2	1 August 2016
Efficacy of NNC0142-0002 in Subjects With Rheumatoid Arthritis (RA)	NCT01181050	Completed	63 participants	Inflammation rheumatoid arthritis	Phase 2	3 October 2016
First-in-man Trial of NNC0142-0002 in Patients With Rheumatoid Arthritis	NCT00927927	Completed	65 participants	Inflammation rheumatoid arthritis	Phase 1	3 October 2016

Abbreviations: NKG2D: natural killer group 2 member D, NCT: national clinical trial, RA: rheumatoid arthritis.

**Table 2 ijms-24-17545-t002:** Different NKG2D-positive lymphocytes involved in autoimmune diseases and observable NKG2D ligands.

Disease	Immune Cell Type	Observed NKG2D Ligands	Biological Effects of NKG2D-NKG2DL	Refs.
SLE	NKG2D^+^CD4^+^ T cell NK cell	MICA sMICB	secretion of IFN, TNF-α, and granzyme promotes expansion	[48,49]
RA	NKG2D^+^CD4^+^ T cell NK cell	MICA sMICA	direct cytotoxicity decreased NKG2D expression	[43,64]
MS	NKG2D^+^CD4^+^ T cell NKG2D^+^CD8^+^ T cell NK cell	sULBP4	promotes migration promotion of CD8^+^T cell motility enhances production of pro-inflammatory cytokines GM-CSF and IFN-γ	[78,81]
T1DM	NKG2D^+^CD8^+^ T cell NK cell	RAE-1	direct cytotoxicity	[89,95,99]
IBD	NKG2D^+^CD4^+^ T cell	MICA	secretion of IFN-γ	[104]
CeD	NKG2D^+^IEL cell	MICA	direct cytotoxicity	[122]

Abbreviations: NKG2D: natural killer group 2 member D, NKG2D-L: NKG2D ligand, SLE: systemic lupus erythematosus, RA: rheumatoid arthritis, MS: multiple sclerosis, T1DM: type I diabetes, IBD: inflammatory bowel disease, CeD: celiac disease, NK: natural killer, IEL: intraepithelial lymphocyte, MICA: MHC class I chain-related protein A, MICB: MHC class I chain-related protein B, sMICA: soluble MICA, sMICB: soluble MICB, sULBP4: soluble UL16-binding protein4, IFN-γ: interferon-γ, TNF-α: tumor necrosis factor-α, GM-CSF: granulocyte-macrophage colony-stimulating factor.

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
