# Peer review of "The Role of NKG2D and Its Ligands in Autoimmune Diseases: New Targets for Immunotherapy"

_ijms, 2023, doi:10.3390/ijms242417545_

Round 1

Reviewer 1 Report

Comments and Suggestions for Authors

this an interesting review compiling recent data in the role of NKG2D/NKG2DL in various autoimmune conditions. A comparison of the different diseases listed in the review, with some comments or a discussion regarding their common or unique feature regarding the impact of NKG2D signalling would have increased the interest of the reader, instead of a classical listing.

Some minor English mistakes indicate that a proofreading will be necessary. 

Otherwise, the manuscript is well illustrated and quite exhaustive.

Comments on the Quality of English Language

minor mistakes remain (eg NKG2D-L are a stress protein that is generally low expressed on normal cells...)

Author Response

Manuscript ID: ijms-2729766

Manuscript title: The role of NKG2D and its ligands in autoimmune diseases: New targets for immunotherapy.

Dear Reviewer:

We are very grateful to your valuable comments on this article, which have been helpful in improving the quality of the manuscript. We have carefully considered and responded to each of the suggestions made by the reviewer. The following is a point-by-point response to the reviewers' comments. We would like to submit this revised manuscript to the “International Journal of Molecular Sciences”, and hope it will be published in the journal.

Looking forward to hearing from you soon.

With kindest regards,

Yours Sincerely,

Yizhou Zou.

Response to reviewer 1

Reviewer 1:

General Comments:

This an interesting review compiling recent data in the role of NKG2D/NKG2DL in various autoimmune conditions. A comparison of the different diseases listed in the review, with some comments or a discussion regarding their common or unique feature regarding the impact of NKG2D signaling would have increased the interest of the reader, instead of a classical listing. Some minor English mistakes indicate that a proofreading will be necessary. Otherwise, the manuscript is well illustrated and quite exhaustive.

Point 1: A comparison of the different diseases listed in the review, with some comments or a discussion regarding their common or unique feature regarding the impact of NKG2D signaling would have increased the interest of the reader, instead of a classical listing.

Response 1: We sincerely appreciate your careful review! Thank you for your recognition of our work. Based on your suggestions, we summarized the NKG2D-positive lymphocytes and their ligands involved in autoimmune diseases, as shown in Table 2 (lines 439-444). In addition, we finally describe Table 2 in the conclusion section (lines 453-457).

Point 2: Some minor English mistakes indicate that a proofreading will be necessary.

Response 2: We appreciate your valuable advice. We apologize for the poor language of the manuscript. Now, we've made changes in both language and readability, and we've also asked native English speakers to correct the language. We sincerely hope that the fluency and language level of the manuscript can be greatly improved. Thank you very much for your valuable comments.

Reviewer 2 Report

Comments and Suggestions for Authors

In their review titled “The role of NKG2D and its ligands in autoimmune diseases: New targets for immunotherapy”, the authors summarize the NK cell features in autoimmune disease development, focusing on the NKG2D-NKG2D Ligand (NKG2D-L) axis. The paper is interesting since the authors describe the role of NK cells that might play in the pathogenesis of autoimmune disease. Furthermore, they also provide the possibilities of therapeutic targeting the NKG2D and NKG2D-L axis for the treatment of autoimmune disease, however, some points need attention.

- The paper does not describe scientifically solidified facts on NK cells in autoimmune disease and is too much narrow focused on NKG2D-NKG2D-L interaction.

-Some abbreviations appear without explanation of the full name. The expression method of abbreviations is not unified.

- Table 1 should be better described in more detail and discussed.

- Please include explanations of abbreviations included in the figure or table below all figure and table descriptions.

- It would be helpful to explain the difference between the roles of NK cells and NKG2D receptors in autoimmune diseases and cancer with more specific examples.

- In Pages 9-10, Chapter 4, it seems necessary to mention the risk of decreased antitumor and antiviral functions of NK cells following the use of NKG2D monoclonal antibody.

- There are some minor typographical errors that should be corrected (points, spaces, grammar-related, mellitus in cursive …). 

Comments on the Quality of English Language

There are some minor typographical errors that should be corrected (points, spaces, grammar-related, mellitus in cursive …).

Author Response

Manuscript ID: ijms-2729766

Manuscript title: The role of NKG2D and its ligands in autoimmune diseases: New targets for immunotherapy.

Dear Reviewer:

We are very grateful to your valuable comments on this article, which have been helpful in improving the quality of the manuscript. We have carefully considered and responded to each of the suggestions made by the reviewer. The following is a point-by-point response to the reviewers' comments. We would like to submit this revised manuscript to the “International Journal of Molecular Sciences”, and hope it will be published in the journal.

Looking forward to hearing from you soon.

With kindest regards,

Yours Sincerely,

Yizhou Zou.

Response to reviewer 2

Reviewer 2:

General Comments:

In their review titled “The role of NKG2D and its ligands in autoimmune diseases: New targets for immunotherapy”, the authors summarize the NK cell features in autoimmune disease development, focusing on the NKG2D-NKG2D Ligand (NKG2D-L) axis. The paper is interesting since the authors describe the role of NK cells that might play in the pathogenesis of autoimmune disease. Furthermore, they also provide the possibilities of therapeutic targeting the NKG2D and NKG2D-L axis for the treatment of autoimmune disease, however, some points need attention.

- The paper does not describe scientifically solidified facts on NK cells in autoimmune disease and is too much narrow focused on NKG2D-NKG2D-L interaction.

-Some abbreviations appear without explanation of the full name. The expression method of abbreviations is not unified.

- Table 1 should be better described in more detail and discussed.

- Please include explanations of abbreviations included in the figure or table below all figure and table descriptions.

- It would be helpful to explain the difference between the roles of NK cells and NKG2D receptors in autoimmune diseases and cancer with more specific examples.

- In Pages 9-10, Chapter 4, it seems necessary to mention the risk of decreased antitumor and antiviral functions of NK cells following the use of NKG2D monoclonal antibody.

- There are some minor typographical errors that should be corrected (points, spaces, grammar-related, mellitus in cursive …).

Point 1: The paper does not describe scientifically solidified facts on NK cells in autoimmune disease and is too much narrow focused on NKG2D-NKG2D-L interaction.

Response 1: Thank you very much for your comments and professional advice, which have helped to improve the academic rigor of our article. NKG2D is one of the important activating receptors on the surface of NK cells, which is expressed in not only NK cells, but also CD8+T cells, γδT cells and a small part of CD4+T cells. However, in autoimmune diseases, the expression of NKG2D is altered. Therefore, we summarized NKG2D-positive lymphocytes and NKG2D ligands involved in autoimmune diseases, and described how the interaction of NKG2D-NKG2DL affects autoimmune diseases. For the role of NK cells in autoimmune diseases, we describe them in SLE, RA, MS, T1DM, and CeD (yellow highlights: lines 185-189, 207-211, 271-273, 303-312, 399-401).

Point 2: Some abbreviations appear without explanation of the full name. The expression method of abbreviations is not unified.

Response 2: Thank you very much for your careful examination. Based on your suggestions, we have standardised the expression of acronyms appearing in the paper and added a summary of acronyms at the end of the paper (lines 464-534).

Point 3: Table 1 should be better described in more detail and discussed.

Response 3: We appreciate your valuable advice. We have added a description of Table 1 to the paper (lines 412-414). For the clinical trials that published papers (NCT02877134 and NCT01203631), we provide a detailed description and discussion (see green highlighted section, lines 369-375).

Point 4: Please include explanations of abbreviations included in the figure or table below all figure and table descriptions.

Response 4: Thank you very much for your careful examination. Based on your suggestion, we have added explanations of the abbreviations present in the charts and tables at the bottom of the charts and tables (lines 65-69, 107-114, 236-238, 355-357, 437, 440-444).

Point 5: It would be helpful to explain the difference between the roles of NK cells and NKG2D receptors in autoimmune diseases and cancer with more specific examples.

Response 5: Thank you very much for your comments and professional advice. Based on your suggestion, we have added a description of the difference between the two roles in the paper (lines 150-167).

Point 6: In Pages 9-10, Chapter 4, it seems necessary to mention the risk of decreased antitumor and antiviral functions of NK cells following the use of NKG2D monoclonal antibody.

Response 6: Thank you very much for your comments and professional advice. These comments helped to improve the academic rigor of our article. Based on your suggestion, we have added a description of this risk to the paper (lines 426-431).

Point 7: There are some minor typographical errors that should be corrected (points, spaces, grammar-related, mellitus in cursive …).

Response 7: Thank you so much for your careful check. We feel so sorry for the mistakes in the manuscript and inconvenience they caused in your reading. For grammatical errors, the manuscript has been thoroughly revised and edited by native English-speaking experts in the hope that it will meet the standards for final publication. For the typographical errors, we have revised them in the paper. Thank you very much for your valuable comments.